# A Probabilistic Perspective on Unlearning and Alignment for Large Language Models

**Yan Scholten, Stephan Günnemann, Leo Schwinn**
Department of Computer Science & Munich Data Science Institute
Technical University of Munich
{y.scholten, s.guennemann, l.schwinn}@tum.de

## Abstract

Comprehensive evaluation of Large Language Models (LLMs) is an open research problem. Existing evaluations rely on *deterministic* point estimates generated via greedy decoding. However, we find that deterministic evaluations fail to capture the whole output distribution of a model, yielding inaccurate estimations of model capabilities. This is particularly problematic in critical contexts such as unlearning and alignment, where precise model evaluations are crucial. To remedy this, we introduce the first formal *probabilistic* evaluation framework for LLMs. Namely, we propose novel metrics with high probability guarantees concerning the output distribution of a model. Our metrics are application-independent and allow practitioners to make more *reliable* estimates about model capabilities before deployment. Our experimental analysis reveals that deterministic evaluations falsely indicate successful unlearning and alignment, whereas our probabilistic evaluations better capture model capabilities. We show how to overcome challenges associated with probabilistic outputs in a case study on unlearning by introducing (1) a novel loss based on entropy optimization, and (2) adaptive temperature scaling. We demonstrate that our approach significantly enhances unlearning in probabilistic settings on recent benchmarks. Overall, our proposed shift from point estimates to probabilistic evaluations of output distributions represents an important step toward comprehensive evaluations of LLMs.[1]

## 1 Introduction

Large Language Models (LLMs) are widely employed across various applications, from chatbots to code generation, relying on outputs generated through **probabilistic** decoding methods such as beam-search and multinominal sampling. Despite their probabilistic deployment, performance evaluations in LLMs predominately rely on **deterministic** point estimates, where outputs are generated through greedy decoding. This raises a critical research question:

> *Are deterministic evaluations adequate for assessing sensitive applications*
> *or do they fall short in capturing the risks associated with probabilistic outputs?*

Current deterministic evaluations may be misaligned with practical usage by overlooking the inherent variability in model outputs. As a result, they could fail to account for both utility and potential risks associated with the model's entire output distribution. Yet, use cases like model alignment and unlearning require precise evaluations to mitigate the risk of harmful usage or privacy non-compliance during deployment. As illustrated in Figure 1, an unlearning algorithm might seem to successfully delete information in a deterministic setting but still leak it probabilistically.

In many scenarios, leakage in even a small fraction of samples (such as revealing social security numbers, passwords, or copyrighted content) can be as problematic as widespread leakage. To address this, we empirically assess whether deterministic methods adequately reflect the risk of information leakage. We find that current deterministic evaluations are insufficient and do not capture practical risks in real-world probabilistic settings, and we propose evaluating the LLM's entire output distribution instead of relying on single-point estimates.

---

[1]Project page: https://www.cs.cit.tum.de/daml/probabilistic-unlearning/

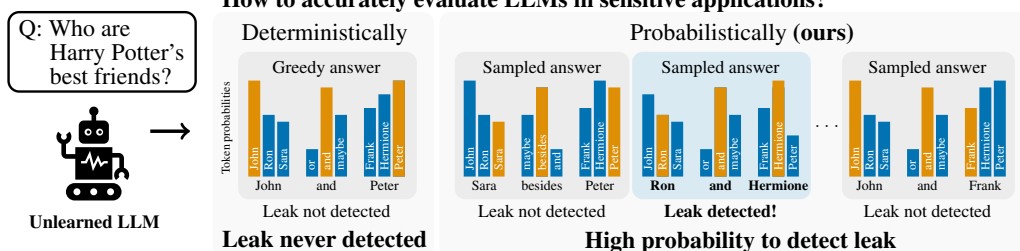

Figure 1: We propose a novel **probabilistic evaluation framework** as a more reliable method for assessing LLM capabilities. Existing evaluations are deterministic and rely on greedy decoding, where the most likely token is selected at each step, producing only a single output per query. Since *in most practical applications LLMs generate outputs probabilistically*, previous evaluation schemes are insufficient: They overlook potential information leaks and falsely suggest successful unlearning. In contrast, in our probabilistic evaluation framework we directly consider the LLM's output distribution by sampling from the token probability distribution at each step to generate multiple sequences. In an empirical study, we show that all state-of-the-art unlearning methods leak information under our probabilistic setting, demonstrating that current deterministic evaluations are insufficient.

Our main contributions are:

- We demonstrate that simple multinominal sampling breaks all state-of-the-art unlearning algorithms and aligned models, retrieving most if not all of the unlearned or toxic information.
- We are the first to formally model LLM evaluations from a probabilistic perspective and thereby capture the practical risk of information leakage more accurately than existing approaches.
- We propose a probabilistic evaluation framework consisting of a suite of principled metrics for comparing LLM output distributions with high-probability guarantees.
- We demonstrate how to reduce information leakage in probabilistic unlearning settings by introducing (1) a novel loss based on entropy optimization, and (2) adaptive temperature scaling.

## 2 RELATED WORK

**Machine Unlearning.** Machine unlearning aims to remove specific information from a model's weights while preserving its overall capabilities (Cao & Yang, 2015). Early works focus on classification tasks (Guo et al., 2020; Golatkar et al., 2020; Tanno et al., 2022; Wang et al., 2023; Pawelczyk et al., 2024). Later works consider more complex scenarios, such as language models for text generation (Jang et al., 2023; Chen & Yang, 2023; Eldan & Russinovich, 2023; Kim et al., 2023; Maini et al., 2024; Sheshadri et al., 2024; Li et al., 2024), which we will focus on. Maini et al. (2024) introduce a synthetic benchmark dataset that allows for controlled learning and unlearning of fictional information. Other works explore broader unlearning contexts, such as removing knowledge about pop culture topics like Harry Potter (Eldan & Russinovich, 2023). Previous algorithms introduce considerable trade-offs between model capability and effectiveness of unlearning, this includes gradient ascent and gradient difference (Liu et al., 2022), Kullback-Leibler minimization, or preference optimization (Rafailov et al., 2024). Zhang et al. (2024) propose negative preference optimization, which shows notable improvements in balancing model capability and unlearning quality.

**Attacks against unlearning.** Towards more accurate evaluations of unlearning, recent studies have explored whether information supposedly removed by unlearning algorithms can be retrieved using extraction attacks. Patil et al. (2024) utilize a logit lens (Geva et al., 2021) approach to analyze hidden states of LLMs to extract unlearned information. Recently, adversarial attacks in the embedding space of LLMs have been proposed to retrieve harmful (Schwinn et al., 2023) and unlearned information (Schwinn et al., 2024). Subsequent works demonstrate that continuous attacks can be used to defend models against such threats (Sheshadri et al., 2024; Xhonneux et al., 2024). Moreover, Lynch et al. (2024) propose a diverse set of methods to robustly evaluate unlearning in LLMs. Beyond extraction attacks, recent studies aim to quantify the degree of memorization in LLMs. Carlini et al. (2023) estimate that these models memorize at least $1\%$ of their training dataset. Schwarzschild et al. (2024) introduce the adversarial compression ratio as a metric that measures the difficulty of eliciting predefined responses with significantly shorter input prompts.

**Certified machine unlearning.** Beyond empirical unlearning methods, first works guarantee exact unlearning (Bourtoule et al., 2021), and approximate unlearning based on differential privacy (Guo et al., 2020; Neel et al., 2021; Ullah et al., 2021; Chien et al., 2022; Zhang et al., 2023) and generalization theory (Sekhari et al., 2021). All of these methods propose adapted training techniques that are aware of the need for later unlearning and consequently require training access. However, such methods are not applicable in settings where models have already been trained on data that needs to be unlearned, and are thereby particularly impracticable for LLMs. In contrast, we investigate unlearning for LLMs after models have been trained on data that needs to be unlearned.

## 3 PRELIMINARIES

**Language models.** Without loss of generality, we model language models as parameterized functions $\pi_\theta : V^* \to \Delta^{|V|^m - 1}$ mapping an input sequence of arbitrary length to a distribution over output sequences of length $m$, where $\theta$ are the model parameters, $V$ denotes a vocabulary, and $\Delta^{|V|^m - 1}$ is the probability simplex in $\mathbb{R}^{|V|^m}$. In other words, for a fixed input sequence $x \in V^*$, $\pi_\theta(x)$ spans a probability distribution over all possible output sequences $V^m$ of length $m$. While we are generally interested in the output distribution $\pi_\theta(x)$, in practice we cannot directly access this distribution since the number of possible output sequences $|V|^m$ quickly outgrows the number of atoms in the observable universe. Instead, we can only access and evaluate the language model sequentially as follows: $\pi_\theta(y_1, \ldots, y_m | x) = \prod_{t=1}^m \pi_\theta(y_t | y_{t-1}, \ldots, y_1, x)$, where $\pi_\theta(y_t | y_{t-1}, \ldots, y_1, x)$ is the conditional probability of token $y_t$ given previous tokens $y_{t-1}, \ldots, y_1$ and input sequence $x$. This represents a challenge: Without any further knowledge about the distribution $\pi_\theta(x)$, practically we can only learn about it via sampling the model's responses for a given input sequence $x$, $Y \sim \pi_\theta(x)$.

**Deterministic evaluation metrics.** Assume we have a perfect oracle to decide if a generated text leaks toxic or sensitive information. We model this using a function $h : V^m \to [0, 1]$ that quantifies how much information got leaked, where $h(s) = 0$ means $s$ does not leak information, and $h(s) = 1$ means complete leakage. For example, $h$ can be binary and indicate if specific data got leaked, or the ROUGE score, which measures the similarity between the model's response and a ground truth.

**Machine unlearning.** The goal of machine unlearning is to remove knowledge from a model while preserving its overall performance. That is, given a model $\pi_\theta$, a forget set $\mathcal{D}_{FG}$, and a retain set $\mathcal{D}_{RT}$, we seek an algorithm to transform the model's parameters $\theta$ such that the response $y$ of the updated model $\pi_{\tilde{\theta}}$ does not answer the queries $x$ for all $(x, y) \in \mathcal{D}_{FG}$ of the forget set. The challenge is that the model's utility should remain high for queries from the retain set $\mathcal{D}_{RT}$ at the same time.

## 4 A COMPREHENSIVE EVALUATION FRAMEWORK FOR LLMS

Current evaluation schemes are insufficient to evaluate LLMs in sensitive applications since they are based on point estimates. To remedy this, we propose a probabilistic evaluation framework. For the sake of clarity, we introduce our framework using the application case of machine unlearning, although our framework generalizes beyond unlearning to other domains as well. First, we properly define four desiderata for machine unlearning that comprehensive evaluations must fulfill:

> **Desiderata for comprehensive machine unlearning evaluations**
>
> **I:** Must quantify the extent of unlearning.
> **II:** Must be efficient to ensure feasibility in practical deployments.
> **III:** Must accurately reflect practical leakage risks (e.g., when sampling from the model) and must detect residual information contained in the unlearned model.
> **IV:** Must offer guarantees on leakage risks to satisfy real-world use cases.

Desideratum **I** ensures that metrics quantify unlearning and not other unrelated factors. **II** addresses the practicality of implementing evaluations in real-world scenarios. **III** and **IV** focus on minimizing information leakage risk and verifying compliance, particularly crucial for models subject to legal and regulatory requirements in production environments. Guided by our desiderata for comprehensive machine unlearning evaluations we introduce our probabilistic evaluation framework, proposing metrics with high-probability guarantees for final evaluations in leakage-sensitive environments, along with a metric to help practitioners assess unlearning quality during development.

## 4.1 METRICS FOR COMPREHENSIVE EVALUATIONS OF OUTPUT DISTRIBUTIONS

Computing metrics with guarantees is challenging especially for LLMs since their output distributions are complex and we cannot make any assumptions about them. We propose to overcome this challenge through (1) Monte Carlo sampling to estimate distribution properties, and by (2) introducing novel metrics with formal guarantees based on distribution-free, non-parametric bounds. Specifically, our metrics are based on concentration bounds that are widely used in the literature, e.g. in the context of probabilistic certifiable robustness (expectation-bounds (Lécuyer et al., 2019; Cohen et al., 2019), CDF-bounds (Kumar et al., 2020), variance-bounds (Schuchardt et al., 2023)).

Let $q$ denote an input prompt and $Y \sim \pi_\theta(q)$ a sequence sampled from the complex distribution that LLMs span over output sequences given $q$. To quantify leakage in probabilistic settings, we compute metrics on the random variable $X = h(Y)$, where $h$ quantifies leakage for a single answer $Y$. Specifically, we first sample $n$ independent realizations $Y_1, \ldots, Y_n$ of $Y$ and measure the extent of leakage $X_i = h(Y_i)$ in each realization. Finally, we compute our probabilistic metrics $M(X_1, \ldots, X_n)$, where $M$ can be replaced by the chosen metric that we introduce in the following. We summarize this procedure in Algorithm 1.

---

**Algorithm 1** Metrics computation

**Require:** Probabilistic metric $M$
1: Sample $n$ answers from LLM $\pi_\theta$
   $Y_1, \ldots, Y_n \sim \pi_\theta(q)$
2: Compute evaluation measure
   $X_i = h(Y_i)$ for $i = 1, \ldots, n$
3: Compute probabilistic metric
   $M(X_1, \ldots, X_n)$

---

We now introduce four probabilistic metrics $\mathbf{M_{bin}}, \mathbf{M_{gen}}, \mathbf{M_\mu}, \mathbf{M_\sigma}$, which require that one specifies a significance level $\alpha \leq \frac{1}{2}$, i.e. our metrics hold with an (arbitrarily high) probability of $1 - \alpha$.

**Binary case.** First we consider binary evaluation metrics $h : V^m \to \{0, 1\}$, where $h(Y) = 1$ means information got leaked. Then $X$ is a Bernoulli random variable with success probability $p$ corresponding to the probability of leaking information. We can upper bound $p$ by sampling from the model's output distribution and by computing a binomial confidence bound: Let $S_n = \sum_{i=1}^n X_i$ count how often information got leaked when sampling from the LLM, where $n$ is the number of Monte-Carlo samples. We propose to compute the following Clopper-Pearson upper confidence bound (Clopper & Pearson, 1934) to quantify information leakage (Proof in Appendix C):

**Metric 1** (Binary leakage bound). *We define the binary metric $M_{bin} \triangleq B(1 - \alpha; S_n + 1, n - S_n)$ where $B(\hat{q}; a, b)$ is the $\hat{q}$th-quantile of the beta distribution with shape parameters $a$ and $b$.*

**Proposition 1.** *With high probability of at least $1 - \alpha$, metric $M_{bin}$ represents an upper bound on the probability that the next sample leaks information, $p \leq M_{bin}$.*

**General case.** Most applications will require more fine-grained metrics for quantifying information leakage. Considering the general case of arbitrary evaluation metrics $h : V^m \to [0, 1]$, we propose to bound the probability $\Pr[X > x]$ that models leak more than a certain threshold $x$. To this end, we bound the CDF $F(x)$ of $X$ with the empirical CDF $F_n(x) = \frac{1}{n} \sum_{i=1}^n \mathbb{1}\{X_i \leq x\}$, which counts how many times at most x% got leaked given $n$ samples. This can be achieved with the Dvoretzky-Kiefer-Wolfowitz (DKW) inequality, which guarantees that the empirical CDF is a close approximation:

$\Pr\left(\sup_{x \in \mathbb{R}} F_n(x) - F(x) > \epsilon\right) \leq e^{-2n\epsilon^2}$ for all $\epsilon \geq \sqrt{\frac{\ln(1/2)}{-2n}}$ (Dvoretzky et al., 1956).

We introduce the following metric to quantify information leakage in general (Proof in Appendix C):

**Metric 2** (General leakage bound). *Given a specified percentage $x \in [0, 1]$ of the information the model should not leak, we define the metric $M_{gen}(x) \triangleq 1 - F_n(x) + \epsilon$ with $\epsilon = \sqrt{\frac{\ln(1/\alpha)}{2n}}$.*

**Proposition 2.** *With high probability of at least $1 - \alpha$, metric $M_{gen}(x)$ upper-bounds the probability that the next sample leaks more than x% of the information, $\Pr(X > x) \leq M_{gen}(x)$ for all $x \in [0, 1]$.*

## 4.2 QUANTIFYING OUTPUT DISTRIBUTIONS WITH MOMENT BOUNDS

Besides bounding the probability of leaking information, we can also quantify information leakage by bounding moments of the output distribution of LLMs. In particular, we propose metrics by bounding moments of the random variable $X = h(Y)$ with high probability using CDF bounds.

**Expectation bounds.** First, we propose to bound the expected secret leakage $\mathbb{E}[X]$ with high probability. Let the points $(\tau_0, \ldots, \tau_K)$ partition the interval $[0, 1]$ into $K$ disjoint intervals, meaning $0 = \tau_0 \leq \tau_1 \leq \ldots \leq \tau_K = 1$. Our metrics are based on the following result (Proof in Appendix C):

**Proposition 3** (Anderson (1969)). *We have $\mathbb{E}[X] \in [\underline{\mu}, \overline{\mu}]$ with high probability of at least $1 - \alpha$ for*

$$\underline{\mu} = 1 - \sum_{i=1}^{K} \delta_{i-1}(F_n(\tau_i) + \epsilon) \quad and \quad \overline{\mu} = 1 - \sum_{i=0}^{K-1} \delta_i(F_n(\tau_i) - \epsilon)$$

*where $F_n(x) = \frac{1}{n} \sum_{i=1}^{n} \mathbb{1}\{X_i \leq x\}$ is the empirical CDF, $\epsilon = \sqrt{\frac{\ln(2/\alpha)}{2n}}$ and $\delta_i = \tau_{i+1} - \tau_i$.*

We can use the upper bound of Proposition 3 to define the following metric:

**Metric 3** (Expectation bound). *We define the metric $M_\mu \triangleq 1 - \sum_{i=0}^{K-1} \delta_i(F_n(\tau_i) - \epsilon)$ that bounds the expected leakage $\mathbb{E}[X]$ of information with high probability of at least $1 - 2\alpha$.*

**Standard deviation bounds.** The second moment-based metric we propose is an upper bound on the standard deviation of $X$. First we compute the bounds $\overline{F}(x) = F_n(x) + \epsilon$ and $\underline{F}(x) = F_n(x) - \epsilon$ on the CDF $F(x)$ via the DKW inequality (Dvoretzky et al., 1956). We then use the bounds on the expectation $\underline{\mu}, \overline{\mu}$ of Proposition 3 to propose the following metric (Proof in Appendix C):

**Metric 4** (Standard deviation bound). *We define the metric $M_\sigma \triangleq \overline{\sigma}$ for*

$$\overline{\sigma}^2 = \eta_{K-1} - \eta_0 \underline{F}(\tau_0) + \sum_{i=1}^{K-1} \delta_i \left[ \text{sign}(\delta_i)\overline{F}(\tau_i) + (1 - \text{sign}(\delta_i))\underline{F}(\tau_i) \right]$$

*where $\eta_i = \max_{\kappa \in \{\tau_i, \tau_{i+1}\}, a \in \{\underline{\mu}, \overline{\mu}\}} (\kappa - a)^2$ for $i \in \{0, \ldots, K-1\}$ and $\delta_i = \eta_{i-1} - \eta_i$.*

**Proposition 4.** *With high probability of at least $1 - \alpha$, metric $M_\sigma(x)$ upper-bounds the standard deviation of $X$, $\sqrt{\text{Var}[X]} \leq M_\sigma$.*

### 4.3 METRICS FOR QUANTIFYING OUTPUT DISTRIBUTIONS DURING MODEL DEVELOPMENT

While metrics with high-probability guarantees on the output distribution of LLMs are critical for final evaluations in leakage-sensitive environments, practitioners also require metrics that are both efficient and easy to compute during development. To meet this need, we introduce the Expectation-Deviation score (**ED score**), which combines expectation and deviation of the distribution of $X$ into a single metric, offering an effective measure of e.g. unlearning quality during model development:

$$S_{ED}(\{X_1, \ldots, X_n\}) = S_{mean} + \rho \cdot S_{sd}$$

where $S_{mean} = \frac{1}{n} \sum_{i=1}^{n} X_i$ is the sample mean and $S_{sd} = \sqrt{\frac{1}{n} \sum_{i=1}^{n} (X_i - S_{mean})^2}$ the sample standard deviation. Here, $\rho$ controls the trade-off between mean and standard deviation, representing an application-dependent parameter that can be adjusted based on the application's risk level. In our unlearning experiments we set $\rho = 2$ to balance the two components.

## 5 DISTRIBUTION UNLEARNING USING ENTROPY OPTIMIZATION AND ADAPTIVE TEMPERATURE SCALING

Existing unlearning methods typically focus on the greedy output of a model's output distribution, $\pi_\theta(x)$, overlooking that the unlearned data may still be embedded in the broader distribution rather than the point estimate. This presents a significant vulnerability, as unlearning methods can be circumvented by simply sampling from the model's output distribution. In addition to introducing improved metrics for evaluating unlearning success from a probabilistic perspective, we propose a novel approach that accounts for output distributions during machine unlearning itself. Our method utilizes entropy optimization and adaptive temperature scaling, which we describe in the following:

**Entropy optimization.** First, our goal is to minimize the entropy of the model's output distribution for forget samples $\mathcal{D}_{FG}$. To this end, we define the following loss function that corresponds to the entropy of the distribution $\pi_\theta(y|y_{t-1}, \ldots, y_1, x)$ over the possibilities $y$ for the next token, given the previous tokens $y_{t-1}, \ldots, y_1$ and the input sequence $x$, averaged over all tokens of the sequence: $\ell_\theta(x, y) = \frac{1}{m} \sum_{t=1}^{m} H(\pi_\theta(y|y_{t-1}, \ldots, y_1, x))$, where $H(q) = -\sum_{i=1}^{|V|} q_i \log q_i$ is the entropy.

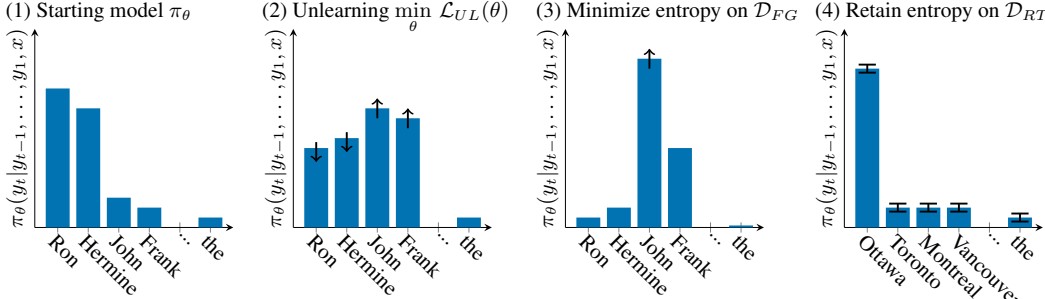

Figure 2: **Entropy optimization:** In this example the model (1) must unlearn the answer to the question "Who are Harry Potter's best friends?" while retaining the answer to the question "What is the capital of Canada?". While minimizing the unlearning loss (2) ensures that the model forgets the sensitive information, our method minimizes the entropy of the model's output distribution for forget samples (3) and retains it on retain samples (4). This allows us to selectively reduce entropy for unlearning-related queries while maintaining entropy on retain samples, effectively reducing the risk of leaking sensitive information under sampling attacks without compromising diversity.

Minimizing the expected loss $\mathbb{E}_{\mathcal{D}_{FG}}[\ell_\theta(x, y)]$ over forget samples $(x, y) \sim \mathcal{D}_{FG}$ will force the model to output sequences with lower variability and thus reduce the risk of leaking sensitive information. While minimizing the entropy of the model's output distribution for forget samples is crucial for unlearning, it is equally important to retain the model's output diversity for retain samples. In practice, this can be achieved by introducing an opposing loss term to slightly maximize the expected loss $\mathbb{E}_{\mathcal{D}_{RT}}[\ell_\theta(x, y)]$ for retain samples $(x, y) \sim \mathcal{D}_{RT}$ with the objective to maintain the model's entropy for retain distributions. Overall, we propose the following entropy optimization loss given a fixed positive entropy weight $\lambda_f > 0$ and (small) negative entropy weight $\lambda_r < 0$:

$$\mathcal{L}_{EO}(\theta) = \mathcal{L}_{UL}(\theta) + \lambda_f \mathbb{E}_{\mathcal{D}_{FG}}[\ell_\theta(x, y)] + \lambda_r \mathbb{E}_{\mathcal{D}_{RT}}[\ell_\theta(x, y)]$$

where $\mathcal{L}_{UL}(\theta)$ denotes an existing unlearning loss, for example the NPO loss (Zhang et al., 2024). By applying a positive entropy weight $\lambda_f$ to forget samples and a negative weight $\lambda_r$ to retain samples we aim to selectively reduce output diversity for unlearning-related queries while preserving variability elsewhere (see visualization in Figure 2). Notably, our entropy optimization method is highly modular and can be applied on top of any existing unlearning method.

**Adaptive temperature scaling.** As we demonstrate in our experiments, entropy optimization is an effective method to decrease the model's entropy for questions related to the forget set while retaining the entropy of the output distribution for unrelated data. This allows us to additionally adjust the temperature of the model adaptively depending on the certainty of the current generation $c(x) = \frac{1}{m} \sum_{t=1}^{m} p(\hat{y}_t | y_{t-1}, \ldots, y_1, x)$, where $p(\hat{y}_t | y_{t-1}, \ldots, y_1, x)$ is the probability of the most likely token $\hat{y}_t$ of the distribution $\pi_\theta(y | y_{t-1}, \ldots, y_1, x)$ over all possible tokens $y$. Specifically, we define a confidence threshold $c_T$ and set the temperature $\tau$ of the model to 0 if the average confidence of the sequence $c(x)$ is over the threshold. This further reduces the risk of information leakage under sampling with no considerable effect on the diversity of the model outputs. Although hard thresholding was sufficient to substantially decrease information leakage with no effect on generation diversity in our experiments, more sophisticated temperature scaling could be applied to further improve the trade-off between diversity and information leakage in the future.

## 6    EXPERIMENTAL EVALUATION

In the following we provide experimental evaluations on recent alignment and unlearning datasets, demonstrating that **existing deterministic evaluations are insufficient** for capturing practical risks. We show that (1) previous methods are prone to significant leakage, and that (2) we can measure the residual information contained in a model more accurately by using our probabilistic evaluation framework (see §4). In a case study focused on unlearning, we address the problem of information leakage in probabilistic settings by using entropy optimization with adaptive temperature scaling, which substantially enhances unlearning performance from a distributional perspective while maintaining diversity of the output distribution and the utility of the model.

We provide detailed descriptions of the corresponding experimental setups and hyperparameters regarding all methods for the unlearning and alignment experiments in Appendix A. In short:

**Experimental setup for unlearning.** We conduct experiments on TOFU (Maini et al., 2024), which consists of 200 fictitious author profiles split into a retain and forget set, where the retain set is used to maintain model capabilities and the forget set is used for unlearning. All TOFU experiments are performed with the Phi-1.5 model (Li et al., 2023). In addition to TOFU, we conduct experiments on the Llama-2-Who-is-Harry-Potter model, which was unlearned to remove any Harry Potter-related knowledge (Eldan & Russinovich, 2023). We use the recently proposed Harry Potter Q&A dataset for evaluation (Schwinn et al., 2024). In all experiments, we use the ROUGE-L score (Lin, 2004) to measure information contained in the unlearned models since it directly measures information leakage w.r.t. a ground truth reference and is widely used in unlearning. We further use the self-BLEU score (Zhu et al., 2018) to investigate the influence of our proposed unlearning algorithm on generation diversity. As baselines we use Gradient Ascent (GA), Gradient Difference (GD) (Liu et al., 2022), RMU (Li et al., 2024), and NPO (Zhang et al., 2024), and combine NPO with entropy optimization and adaptive temperature scaling for our approach since it is the current state-of-the-art.

**Experimental setup for alignment.** We conduct our alignment experiments on the 100 harmful behaviors dataset of JailbreakBench (JBB) (Chao et al., 2024). Toxicity scores are derived from the Harmbench toxicity classifier (Mazeika et al., 2024), which provides the probability of an answer being rated as toxic. We conduct our evaluations on Phi-1.5 (Li et al., 2023), Vicuna-7b-1.5 (Chiang et al., 2023), and Mistral-7b-instruct-v0.3 (Jiang et al., 2023).

## 6.1 Unlearning case study: Improving evaluations in probabilistic settings

Existing metrics used to measure unlearning quality in LLMs already fulfill desiderata **I** and **II**, i.e., they quantify the extent of unlearning and are efficient to compute. In the following case study on unlearning, we show that point-wise evaluations do not satisfy the remaining desiderata and are thus insufficient for capturing practical risks. To address their limitations and satisfy all desiderata, we use the metrics introduced in our probabilistic evaluation framework (§4), which address desiderata **III** and **IV** and better capture the risk of information leakage in probabilistic settings.

**Unlearning on Harry Potter Q&A.** Figure 3 (a) compares unlearning evaluations conducted either with (deterministic) greedy decoding or probabilistic sampling given the Llama-2-Who-is-Harry-Potter model. We adopt the approach of Schwinn et al. (2024) and define information as leaked if a generated answer contains the relevant keyword for a given question. This binary leakage (either present or absent) allows us to apply our binary leakage bound ($\mathbf{M_{bin}}$) to quantify the extent of information leakage. While point-wise evaluations wrongly indicate that no information is contained in the model after unlearning, in our experiment, simply sampling from the model's output distribution reveals that the model still leaks information (i.e., generates correct responses to the questions). Thus, the deterministic evaluation violates desiderata **III** and **IV**, underestimating the leakage risk and providing no guarantee that the model does not leak information in a deployment scenario. In contrast, our probabilistic binary leakage bound gives a more accurate estimate of the residual information still contained in the model (**III**) and provides a high probability guarantee (**IV**).

**Unlearning on TOFU.** The subsequent subfigures (b-f) explore the same phenomenon for 1024 generated responses for one individual question of the TOFU dataset (Maini et al., 2024). In (b-c), we compare leakage of different unlearning methods for this question for both deterministic and probabilistic evaluations. Although the paired unlearning methods exhibit identical leakage under greedy decoding (as indicated by the bold dashed line), their distributions show substantial differences. This demonstrates that models with identical deterministic evaluation metrics can still behave differently during sampling, supporting our finding that deterministic metrics alone are insufficient. In (d), we compute the general leakage bound ($\mathbf{M_{gen}}$), which highlights that NPO exhibits a considerable leakage risk. In Figure 3 (e) we compare the sample estimate $\mu$ and its upper bound $\overline{\mu}$ on the expected leakage $\mathbb{E}[X]$ for different sample sizes, and Figure 3 (f) shows a similar comparison for the standard deviation. The empirical estimates converge quickly with an increasing number of samples in practice, allowing for precise and efficient estimates. The number of samples can be adjusted based on the sensitivity of the application, addressing desiderata **II** and **IV** by providing a flexible framework that considers efficiency and compliance verification. Similar to the Harry Potter Q&A, our probabilistic framework reveals considerable residual information after unlearning.

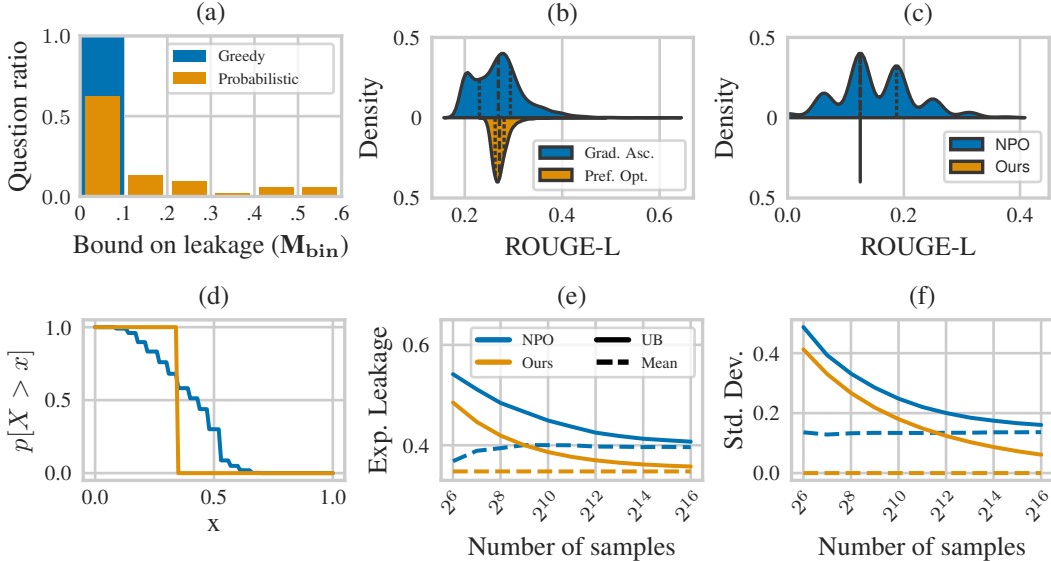

Figure 3: Our results demonstrate that deterministic evaluations fail to detect residual information still contained after unlearning, whereas our probabilistic metrics provide more comprehensive evaluations: (a) Binary leakage bound ($\mathbf{M_{bin}}$) for questions of the Harry Potter Q&A. While greedy decoding indicates successful unlearning, our probabilistic perspective reveals that for 38% of the questions the upper bound on the expected leakage is larger than 10%. (b-c) ROUGE-L score of 1024 generated responses from a single question of the TOFU dataset. The bold dashed line indicates the ROUGE-L score of greedy decoding. The second row contains results for NPO and our proposed unlearning algorithm for a question-answer pair of the TOFU forget set. (d) General leakage bound ($\mathbf{M_{gen}}$) illustrating differences in information leakage between NPO and our approach for different levels of leakage $x$. (e-f) Expectation bound ($\mathbf{M_\mu}$) and standard deviation bound ($\mathbf{M_\sigma}$).

**Extended analysis.** We provide further analysis on the TOFU dataset in Table 1. For unlearning methods GA and GD, the empirical mean matches the ROUGE-L score obtained from greedy decoding, indicating that the point-wise evaluation correctly approximates the leakage risk. However, we observe considerable standard deviation for both methods, indicating substantial leakage. Our proposed ED (Expectation-Deviation) score (§4.3) condenses empirical mean and standard deviation into a single value, offering a direct estimate of the leakage risk. This score provides a practical alternative to metrics with high probability guarantees while remaining more accurate than deterministic evaluations.

Table 1: Comparison of deterministic (det.) and probabilistic (prob.) evaluations on the TOFU dataset (90/10 split). While the point-wise metric already indicates good unlearning, our metrics reveal that their distributions still encode the information.

| | Metric ($\downarrow$) | RMU | GD | GA | NPO | **Ours** |
|---|---|---|---|---|---|---|
| Det. | ROUGE-L | 0.70 | 0.33 | 0.32 | 0.22 | **0.20** |
| Prob. | ED Score | 0.81 | 0.42 | 0.41 | 0.34 | **0.20** |
| | - Mean | 0.60 | 0.32 | 0.31 | 0.21 | **0.20** |
| | - Std. Dev. | 0.10 | 0.05 | 0.05 | 0.06 | **0.00** |

### 6.1.1 THE EFFECT OF ENTROPY REGULARIZATION ON LLM UNLEARNING

To mitigate the leakage risk during sampling, we introduce entropy optimization to selectively decrease the model's entropy on the forget set. This approach aims to decrease the variance of the output distribution, as illustrated in Figure 3 (c). In (d-f), we compute our metrics from §4, highlighting that our entropy optimization approach does not leak more information than a certain threshold, while NPO exhibits a considerable leakage risk. In Figure 4 (a) we further demonstrate the effects of the forget entropy regularization parameter $\lambda_f$ on two TOFU dataset splits (90/10 and 95/5). As we increase the regularization strength, the diversity for unlearning-related queries approaches zero, eliminating the risk of information leakage during sampling. An alternative approach could be lowering the softmax temperature $\tau$ to reduce output diversity. As $\tau$ approaches 0, sampling converges to greedy generation. Figure 4 (b) shows that lower $\tau$ reduces the standard deviation of ROUGE-L scores, indicating less diversity. However, this affects both unlearning-related and unrelated tasks indiscriminately. The next section shows that our approach better preserves diversity.

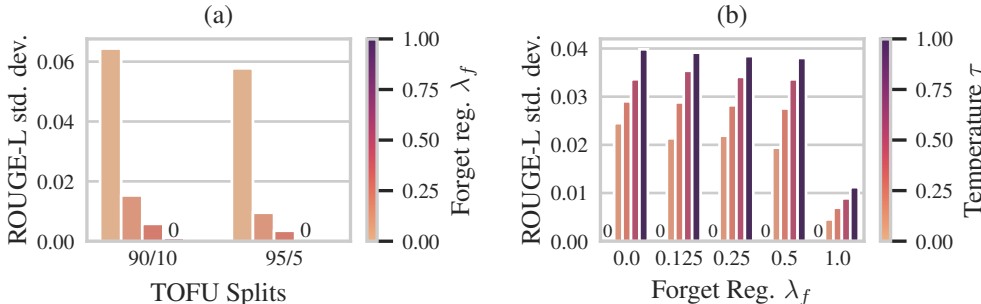

Figure 4: (a) Effect of forget entropy regularization weight $\lambda_f$ on the standard deviation of the leakage distribution. Stronger regularization decreases the probability of leaking information. (b) Decreasing temperature $\tau$ also decreases model leakage, but also results in lower output diversity.

### 6.1.2 MAINTAINING OUTPUT DIVERSITY AND MODEL UTILITY IN LLM UNLEARNING

Entropy optimization effectively reduces information leakage in our experiments. At the same time, unlearning methods should not negatively affect other properties of the model, such as output diversity, model confidence, and overall utility. We investigate these metrics using the *Real Authors* and *World Facts* datasets of TOFU, which were not used during training.

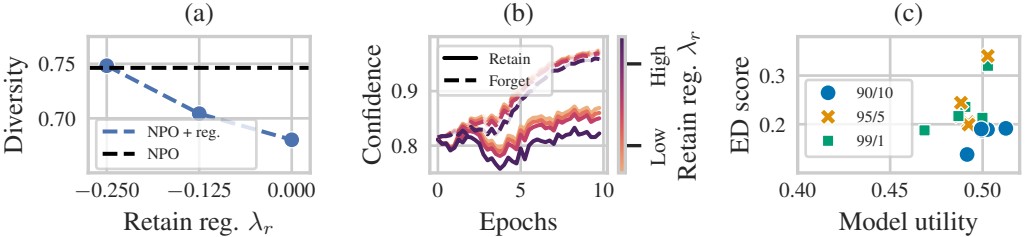

Figure 5: Ablation studies for our proposed entropy optimization approach: (a) Negative effects on output diversity can be mitigated through a negatively weighted ($\lambda_r$) entropy loss. (b) Token confidence on the forget set considerably increases during training, remaining largely the same on the retain set. This allows us to decrease information leakage while maintaining output diversity for unrelated tasks. (c) Models trained with random entropy regularization parameters. We observe no relation between the magnitude of regularization and model utility in our experiments.

**(a) Diversity.** Figure 5 (a) shows the impact of the retain entropy regularization coefficient $\lambda_r$ on output diversity (i.e., $1 - $ self-BLEU) for $\lambda_f = 1$ on the 99/1 split. The final score is obtained by averaging scores across all questions of the dataset and ranges from $0$ (identical outputs) to $1$ (no similarity). The dashed line represents an NPO-unlearned model without entropy regularization, while the blue line shows the entropy-regularized NPO. As $\lambda_r$ decreases (becomes more negative), diversity improves, surpassing the baseline NPO model. This suggests that regularizing entropy on the retain set successfully prevents diversity degradation on datasets unrelated to the forget objective.

**(b) Training confidence trajectories.** Figure 5 (b) illustrates the model's confidence over training epochs for both retain and forget sets. The solid lines represent the retain set, while the dashed lines show the forget set. Multiple trajectories likely represent different experimental conditions or hyperparameter settings. We observe that confidence generally increases over epochs for both sets, with the retain set typically maintaining higher confidence. The trajectories indicate that the model can differentiate between retain and forget information while learning.

**(c) Impact on unlearning and model utility:** Figure 5 (c) compares the ED score against model utility for different data split ratios of the TOFU dataset (90/10, 95/5, 99/1). Each point represents an NPO-unlearned model with random regularization parameters $\lambda_f \in [0, 1]$. In our experiments, the impact of entropy regularization on model utility is minor, with regularized models achieving higher utility than standard NPO in some cases (see Appendix B). Overall, our proposed entropy regularization approach can achieve a nuanced balance between unlearning robustness, output diversity, and overall model utility. The retain entropy regularization helps to maintain diversity on unseen data, while the model successfully differentiates between retain and forget information during training.

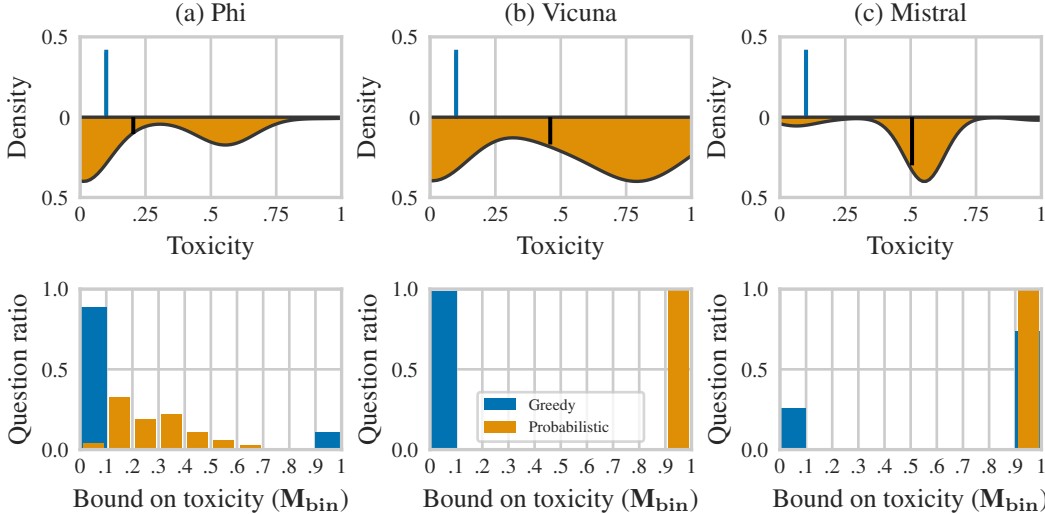

Figure 6: Probabilistic evaluation results for all toxic queries of the JailbreakBench (JBB) dataset. In the first row, we show the toxicity score of 1024 generated responses from a single query of the JBB dataset. The bold black line indicates the mean toxicity value of the probabilistic evaluation, whereas the bold blue line shows the toxicity score of one greedy evaluation for the same question. The expected toxicity value under probabilistic evaluation is consistently higher. The second row shows the binary leakage bound ($\mathbf{M_{bin}}$). While greedy decoding generally indicates that the models are robust, our probabilistic perspective reveals that all models are not robust under sampling.

## 6.2 BEYOND LLM UNLEARNING: PREVIOUS ALIGNMENT EVALUATIONS DO NOT CAPTURE PRACTICAL RISKS

Our probabilistic evaluations only require evaluation measures for single outputs and can be applied seamlessly across various contexts. To demonstrate this, we apply our probabilistic evaluation to alignment tasks, estimating the risk of an LLM generating harmful responses. In the first row of Figure 6 we visualize the fraction of toxic answers among 1024 generated responses for a specific query from the JBB dataset. This is compared to the toxicity observed under deterministic evaluation using greedy decoding. Across all models, average toxicity measured via sampling significantly exceeds that observed through greedy decoding. In the second row, we present the binary leakage bound ($\mathbf{M_{bin}}$) for the full JBB dataset. Results consistently show that greedy decoding underestimates model toxicity, underscoring the limitations of deterministic evaluations in high-stakes applications.

**Limitations.** While our proposed probabilistic evaluation framework approach offers substantial improvements over deterministic point-wise evaluations, it still cannot assess the entire output distribution of LLMs holistically for any possible input. Due to computational constraints, we instead analyze the output distributions of a model for fixed inputs. Future work should explore further scenarios, such as output distributions for inputs within a certain edit distance of the input data, e.g. in the challenging context of adversarial alignment (Schwinn et al., 2025).

## 7 CONCLUSION

We introduce a probabilistic perspective on LLM evaluations and propose a novel framework to directly assess the output distribution of a model. Our proposed perspective shift from single point estimates towards evaluating entire output distributions offers significant potential for the field of unlearning and can be directly used for evaluating a variety of sensitive applications beyond unlearning, such as measuring toxicity and mitigating undesired biases in model outputs. Furthermore, our framework lays the groundwork for developing metrics for quantifying leakage in distributions beyond text, extending to generative models in image, audio, and other modalities. Our work represents an important contribution towards comprehensive LLM evaluations and provides a foundation for future research in this area, such as investigating model utility from a probabilistic perspective.

BROADER IMPACT

Our work highlights the limitations of current LLM evaluations being conducted in a deterministic manner. By introducing a probabilistic evaluation framework, we enable more accurate assessments of model behavior and potential risks. This approach could lead to improved safety and reliability in AI systems, more effective unlearning techniques enhancing privacy protection, and better alignment of AI models. Additionally, our methods could reveal previously unknown vulnerabilities in existing models. Overall, this research contributes to more accurate evaluations of generative models.

REPRODUCIBILITY STATEMENT

We ensure reproducibility by providing detailed documentation of all hyperparameters, model architectures, and experimental setups in Appendix A. All datasets and architectures used in this paper are publicly available, and our implementation is accessible via the following project page: https://www.cs.cit.tum.de/daml/probabilistic-unlearning/.

ACKNOWLEDGMENTS

The authors want to thank Jan Schuchardt for valuable feedback on the manuscript, and Mato Gudelj for providing an initial code base. This work has been funded by the DAAD program Konrad Zuse Schools of Excellence in Artificial Intelligence (sponsored by the Federal Ministry of Education and Research). The authors of this work take full responsibility for its content.

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

## A    EXPERIMENTAL SETUP

**Hardware details.** All experiments are conducted on a NVIDIA A100 GPU with 40GB of memory.

### A.1    EXPERIMENTAL SETUP FOR LLM UNLEARNING EXPERIMENTS

**Datasets and models.** We use two recent unlearning benchmarks for our evaluations. We conduct experiments on TOFU, which consists of 200 fictitious author profiles (Maini et al., 2024). These profiles are split into a retain and forget set, where the retain set is used to maintain model capabilities and the forget set is used for unlearning. Additionally, each profile is divided into multiple question-answer pairs. TOFU provides three different unlearning splits where 99, 95, or 90 percent of the data is used as retain set and the remainder as forget set. For measuring model utility after unlearning, TOFU additionally provides the *Real Authors* and *World Facts* datasets. All TOFU experiments are performed with the Phi-1.5 model (Li et al., 2023). In addition to TOFU, we conduct experiments on the Llama-2-Who-is-Harry-Potter model, which was unlearned to remove any Harry Potter-related knowledge (Eldan & Russinovich, 2023). We use the recently proposed Harry Potter Q&A for evaluation (Schwinn et al., 2024). This dataset consists of pairs of questions and relevant keywords, allowing for the detection of information leakage through keyword matching.

**Baseline metrics.** In all experiments, we use ROUGE-L as a deterministic metric to measure information contained in the model after unlearning. ROUGE-L computes a statistic based on the longest common subsequence between two strings (Lin, 2004). Additionally, we use the ROUGE-L score obtained from multiple sampled responses to compute probabilistic metrics, such as bounds, mean, standard deviation, and the expectation-deviation (ED) score. Note that our framework (§4) can be applied to all deterministic metrics, such as perplexity or forget quality. We chose ROUGE-L as it directly measures information leakage with respect to a ground truth reference and is widely used in the unlearning domain. Throughout the manuscript, we use information leakage to refer to the magnitude of the ROUGE-L score, where a high score indicates high information leakage. We use the model utility score as described in TOFU to measure the generation quality of a given model Maini et al. (2024). We additionally employ the self-BLEU score (Zhu et al., 2018), which computes BLEU scores (Papineni et al., 2002) between generated samples and allows us to investigate the influence of our proposed unlearning algorithm on generation diversity.

**Unlearning methods.** We use Gradient Ascent (GA), Gradient Difference (GD) (Liu et al., 2022), RMU (Li et al., 2024), and NPO (Zhang et al., 2024) for a diverse selection of unlearning baselines and combine NPO with entropy optimization and adaptive temperature scaling for our approach since it is the current state-of-the-art.

**Hyperparameters.** For all unlearning algorithms we use a learning rate of $1e-5$ with a cosine learning rate schedule with warmup ratio of $0.1$, batch size of $32$, and weight decay of $0.01$. For NPO we set $\beta_{NPO} = 0.05$. We use $10$ training epochs for all experiments as in (Maini et al., 2024). For probabilistic evaluations we sample $n = 1024$ model generations for every experiment if not stated otherwise. Probabilistic guarantees are calculated with a high probability guarantee of $\alpha = 0.01$. We set the adaptive temperature scaling threshold $c_T = 0.9$ for all experiments. This was done as the average confidence of all models remained considerably below $0.9$ during training. In our experiments, adaptive temperature thresholding has a negligible effect on the diversity of the model outputs using this threshold (see Section 6.1.2). In the unlearning setting, for each generation we sample sequences that are 64 tokens long. For all generations we used top-p sampling with $p = 0.9$.

### A.2    EXPERIMENTAL SETUP FOR LLM ALIGNMENT EXPERIMENTS

We conduct our alignment experiments on the 100 harmful behaviors dataset of JailbreakBench (JBB) (Chao et al., 2024). Toxicity scores are derived from the Harmbench toxicity classifier (Mazeika et al., 2024), which provides the probability of an answer being rated as toxic. We conduct our evaluations on Phi-1.5 (Li et al., 2023), Vicuna-7b-1.5 (Chiang et al., 2023), and Mistral-7b-instruct-v0.3 (Jiang et al., 2023). For each generation, we sample sequences that are 128 tokens long. For all generations we used top-p sampling with $p = 0.9$.

## B   EFFECT OF ENTROPY OPTIMIZATION ON MODEL UTILITY IN UNLEARNING

In Table 2 we summarize the results of Figure 5 (a) and Figure 5 (c).

Table 2: Utility of unlearned models using the proposed unlearning method for different retain/forget splits of the TOFU dataset and varying choices of entropy regularization. We observe no consistent negative effect of entropy regularization on the utility of an unlearned model.

| $\lambda_f = 1$ | | $\lambda_r = 0$ | | | |
|---|---|---|---|---|---|
| $\lambda_r$ | Utility$_{99}$ | $\lambda_f$ | Utility$_{90}$ | Utility$_{95}$ | Utility$_{99}$ |
| 0.000 | 0.496 | 0.000 | 0.503 | 0.503 | 0.499 |
| -0.125 | 0.509 | 0.125 | 0.488 | 0.491 | 0.503 |
| -0.250 | 0.504 | 0.250 | 0.488 | 0.487 | 0.512 |
| -0.500 | 0.490 | 0.500 | 0.492 | 0.500 | 0.500 |
| | | 1.000 | 0.492 | 0.469 | 0.492 |

## C   METRIC GUARANTEE PROOFS

Note that confidence intervals have two bounds that share a significance level of $\alpha$, meaning each bound uses a significance level of $\alpha/2$. Consequently, since we propose metrics based on one bound only, our bounds can make use of the full significance level $\alpha$.

Recall the definition of the Clopper-Pearson confidence interval (Clopper & Pearson, 1934):

$$B\left(\frac{\alpha}{2}; S_n, n - S_n + 1\right) \leq p \leq B\left(1 - \frac{\alpha}{2}; S_n + 1, n - S_n\right)$$

where $B(\hat{q}; a, b)$ is the $\hat{q}$th-quantile of the beta distribution with shape parameters $a$ and $b$. We propose an unlearning metric based on the conservative Clopper-Pearson confidence bound as follows:

**Metric 1** (Binary leakage bound). *We define the binary metric $M_{bin} \triangleq B(1 - \alpha; S_n + 1, n - S_n)$ where $B(\hat{q}; a, b)$ is the $\hat{q}$th-quantile of the beta distribution with shape parameters $a$ and $b$.*

**Proposition 1.** *With high probability of at least $1 - \alpha$, metric $M_{bin}$ represents an upper bound on the probability that the next sample leaks information, $p \leq M_{bin}$.*

*Proof.* The statement follows directly from the definition of the Clopper-Pearson confidence intervals (Clopper & Pearson, 1934). □

**Metric 2** (General leakage bound). *Given a specified percentage $x \in [0, 1]$ of the information the model should not leak, we define the metric $M_{gen}(x) \triangleq 1 - F_n(x) + \epsilon$ with $\epsilon = \sqrt{\frac{\ln(1/\alpha)}{2n}}$.*

**Proposition 2.** *With high probability of at least $1 - \alpha$, metric $M_2(x)$ upper-bounds the probability that the next sample leaks more than $x$% of the secret, $\Pr(X > x) \leq M_2(x)$ for all $x \in [0, 1]$.*

*Proof.* The Dvoretzky-Kiefer-Wolfowitz inequality guarantees

$$\Pr\left(\sup_{x \in \mathbb{R}} F_n(x) - F(x) > \epsilon\right) \leq e^{-2n\epsilon^2} \qquad \text{for all} \qquad \epsilon \geq \sqrt{\frac{\ln 1/2}{-2n}}$$

Choosing $\epsilon = \sqrt{\frac{\ln(1/\alpha)}{2n}}$ for $\alpha \leq \frac{1}{2}$ we have:

$$\Pr\left(\sup_{x \in \mathbb{R}} F_n(x) - F(x) > \epsilon\right) \leq \alpha$$
$$\Leftrightarrow \Pr\left(F_n(x) - F(x) > \epsilon\right) \leq \alpha \qquad \forall x \in \mathbb{R}$$
$$\Leftrightarrow \Pr\left(F_n(x) - \epsilon > F(x)\right) \leq \alpha \qquad \forall x \in \mathbb{R}$$
$$\Leftrightarrow 1 - \Pr\left(F_n(x) - \epsilon > F(x)\right) \geq 1 - \alpha \qquad \forall x \in \mathbb{R}$$
$$\Leftrightarrow \Pr\left(F_n(x) - \epsilon \leq F(x)\right) \geq 1 - \alpha \qquad \forall x \in \mathbb{R}$$
$$\Leftrightarrow \Pr\left(1 - F_n(x) + \epsilon \geq 1 - F(x)\right) \geq 1 - \alpha \qquad \forall x \in \mathbb{R}$$

We can use the Dvoretzky-Kiefer-Wolfowitz inequality to construct a simultaneous confidence band:

$$p(X > x) \in [1 - F_n(x) - \epsilon, 1 - F_n(x) + \epsilon] \qquad \forall x \in \mathbb{R}$$

where $\epsilon = \sqrt{\frac{\ln(2/\alpha)}{2n}}$. This follows directly from the two-sided DKW inequality:

$$\Pr[\sup_x |F_n(x) - F(x)| > \epsilon] \le \alpha \quad \text{for} \quad \alpha = 2e^{-2n\epsilon^2}$$

$\square$

Note that if in practice we have a fixed $\epsilon$ for a significance level $\alpha$ (for example if we have to guarantee tight bounds), then we can exactly quantify the number of Monte Carlo samples needed: $\alpha = 2e^{-n\epsilon^2} \Leftrightarrow n = \frac{1}{\epsilon^2} \ln\left(\sqrt{\frac{1}{\alpha}}\right)$.

**Metric 3** (Expectation bound). *We define the metric $M_\mu \triangleq 1 - \sum_{i=0}^{K-1} \delta_i(F_n(\tau_i) - \epsilon)$ that bounds the expected leakage $\mathbb{E}[X]$ of information with high probability of at least $1 - 2\alpha$.*

**Proposition 3** (Anderson (1969)). *We have $\mathbb{E}[X] \in [\underline{\mu}, \overline{\mu}]$ with high probability of at least $1 - \alpha$ for*

$$\underline{\mu} = 1 - \sum_{i=1}^{K} \delta_{i-1}(F_n(\tau_i) + \epsilon) \quad \text{and} \quad \overline{\mu} = 1 - \sum_{i=0}^{K-1} \delta_i(F_n(\tau_i) - \epsilon)$$

*where $F_n(x) = \frac{1}{n} \sum_{i=1}^{n} \mathbb{1}\{X_i \le x\}$ is the empirical CDF, $\epsilon = \sqrt{\frac{\ln(2/\alpha)}{2n}}$ and $\delta_i = \tau_{i+1} - \tau_i$.*

*Proof.* We exploit the relation between the CDF and the expectation: $\mathbb{E}[X] = 1 - \int_0^1 F(x)dx$. We have

$$\mathbb{E}[X] = 1 - \int_0^1 F(x)\,dx$$
$$\overset{(1)}{\le} 1 - \sum_{i=0}^{K-1} (\tau_{i+1} - \tau_i)F(\tau_i)$$
$$\overset{(2)}{\le} 1 - \sum_{i=0}^{K-1} (\tau_{i+1} - \tau_i)(F_n(\tau_i) - \epsilon)$$
$$= 1 - \underbrace{\sum_{i=0}^{K-1} \delta_i(F_n(\tau_i) - \epsilon)}_{\overline{\mu}}$$

where inequality (1) holds by lower-bounding the integral with the left Riemann sum, which is a lower bound of the integral since the CDF is monotonically increasing. The second inequality (2) holds due to the Dvoretzky-Kiefer-Wolfowitz inequality.

The lower bound follows analogously:

$$\mathbb{E}[X] = 1 - \int_0^1 F(x)\,dx$$
$$\overset{(1)}{\ge} 1 - \sum_{i=1}^{K} (\tau_i - \tau_{i-1})F(\tau_i)$$
$$\overset{(2)}{\ge} 1 - \sum_{i=1}^{K} (\tau_i - \tau_{i-1})(F_n(\tau_i) + \epsilon)$$
$$= 1 - \underbrace{\sum_{i=1}^{K} \delta_{i-1}(F_n(\tau_i) + \epsilon)}_{\underline{\mu}}$$

where inequality (1) holds by upper-bounding the integral with the right Riemann sum, which is an upper bound of the integral since the CDF is monotonically increasing. The second inequality (2) holds due to the Dvoretzky-Kiefer-Wolfowitz inequality again. □

Following the variance bounds introduced in (Schuchardt et al., 2023) we propose the following bound on the standard deviation as unlearning metric:

**Metric 4** (Standard deviation bound). *We define the metric $M_\sigma \triangleq \overline{\sigma}$ for*

$$\overline{\sigma}^2 = \eta_{K-1} - \eta_0 \underline{F}(\tau_0) + \sum_{i=1}^{K-1} \delta_i \left[ \text{sign}(\delta_i)\overline{F}(\tau_i) + (1 - \text{sign}(\delta_i))\underline{F}(\tau_i) \right]$$

*where $\eta_i = \max_{\kappa \in \{\tau_i, \tau_{i+1}\}, a \in \{\underline{\mu}, \overline{\mu}\}} (\kappa - a)^2$ for $i \in \{0, \ldots, K-1\}$ and $\delta_i = \eta_{i-1} - \eta_i$.*

**Proposition 4.** *With high probability of at least $1 - \alpha$, metric $M_\sigma(x)$ upper-bounds the standard deviation of $X$, $\sqrt{\text{Var}[X]} \leq M_\sigma$.*

*Proof.* We have $\text{Var}[X] = \mathbb{E}[(X - \mathbb{E}[X])^2] = \int_0^1 (x - \mathbb{E}[X])^2 f_X(x)\, dx$

$$= \sum_{i=0}^{K-1} \int_{\tau_i}^{\tau_{i+1}} (x - \mathbb{E}[X])^2 f_X(x)\, dx$$

$$= \sum_{i=0}^{K-1} \int_{\tau_i}^{\tau_{i+1}} (x - \mathbb{E}[X])^2 f_X(x)\, dx$$

$$\leq \sum_{i=0}^{K-1} \eta_i \int_{\tau_i}^{\tau_{i+1}} f_X(x)\, dx \qquad \text{for} \quad \eta_i = \max_{\substack{\kappa \in \{\tau_i, \tau_{i+1}\} \\ a \in \{\underline{\mu}, \overline{\mu}\}}} (\kappa - a)^2$$

$$= \sum_{i=0}^{K-1} \eta_i (F(\tau_{i+1}) - F(\tau_i))$$

$$= \eta_{K-1} - \eta_0 F(\tau_0) + \sum_{i=1}^{K-1} \delta_i F(\tau_i) \qquad \text{for} \quad \delta_i = \eta_{i-1} - \eta_i$$

$$\leq \underbrace{\eta_{K-1} - \eta_0 \underline{F}(\tau_0) + \sum_{i=1}^{K-1} \delta_i \left[ \text{sign}(\delta_i)\overline{F}(\tau_i) + (1 - \text{sign}(\delta_i))\underline{F}(\tau_i) \right]}_{\overline{\sigma}^2}$$

From $\text{Var}[X] \leq \overline{\sigma}^2$ follows $\sqrt{\text{Var}[X]} \leq \overline{\sigma}$ since the square root is monotonically increasing.

□

