# OpenReview forum: "A Probabilistic Perspective on Unlearning and Alignment for Large Language Models"
_ICLR.cc/2025/Conference — ICLR 2025 Oral_

### Official Review · Reviewer_tRL6 · 2024-10-26

**Soundness:** 3
**Presentation:** 2
**Contribution:** 3
**Rating:** 6
**Confidence:** 3

**Summary:**

This work introduces a probablistic perspective for LLM unlearning evaluation. Instead of relying on deterministic greedy decoding in existing evaluation methods, this work takes a probablistic framework and derive metrics considering the high-probable output distributions. The proposed metric demonstrates the limitations of previous methods for their lack of identifying false unlearning. Moreover, a novel loss based on entropy optimization and adaptive temperature scaling are introduced to improve model unlearning.

**Strengths:**

- Designing good evaluation metrics is important for a reserach direction such as unlearning, and this work indicates a limitation of existing metric and correspondingly proposes improved metrics.
- The proposed metrics and methods are shown to be effective for two recent unlearning benchmarks.

**Weaknesses:**

- There is a lack of algorithmic description on how the proposed metrics are calculated, without which readers who lack certain statistical machine learning knowledge or who want to implement the metrics would find it difficult to understand and apply the proposed metrics.
- The proposed metrics are only tested for the unlearning case, which surely is indeed a well-suited scenario. Nevertheless, it would nice if it can be extended to more use cases, such factuality, to verify the effectiveness of the metrics.

**Questions:**

- For entropy optimization, I'm not sure about the intuition to minimize the entropy on Dfg. Wouldn't this lead the model to be confident on a different answer, which I think might be a strange thing to enforce.
- It would be interesting to how unlearning (as well as the proposed optimization methods) affects the model's general ability.

---

> ### Author Response · Authors · 2024-11-19
> **Response to Reviewer tRL6**
>
> Thank you for your review!
>
> **Regarding our proposed metrics.** Please note that all our metrics can be calculated using a single mathematical formula. To make it easier for the community to use our metrics, we will provide our code as a GitHub repository and our framework as a pip library after acceptance. We have already attached the code used to conduct the experiments in the paper as supplemental material.
>
> **Regarding additional use-cases.** We added an additional evaluation on model alignment to our work. We hope the additional experiments and explanations in Appendix B clarify how our approach can be applied to tasks beyond unlearning and strengthen the contribution of the paper. Thank you for pointing out this opportunity for improvement!
>
> **Regarding entropy optimization (Q1).** We conducted an ablation study to investigate if entropy optimization has a negative impact on model properties. In Figure 5 (a) we show that by choosing suitable values for $\lambda_r$, entropy optimization does not affect the diversity of model generations on unseen utility datasets. In Figure 5 (b) we illustrate that the confidence / entropy on the retain set (data unrelated to the unlearning task) remains stable throughout training.
>
> **Regarding the effect on the model's general utility (Q2).**  For the experiments performed in the paper, model utility was not substantially affected by unlearning. In Figure 5 (c) we demonstrate that entropy optimization also does not negatively affect the utility of the model. We will include a table that summarizes the effect of the different unlearning techniques on model utility in the final version of the paper.
>
> We hope that we could address all your questions to your satisfaction. Please let us know if you have any additional comments or questions.

---

> ### Comment · Reviewer_tRL6 · 2024-11-23
> **Response**
>
> Thanks for your response which addresses some of my concerns. And thanks for the plan of code releasing.
> If I'm not misunderstanding, the estimation is mainly the average of N independent sampling (plus 2 times standard deviation)? I still feel that a clearer procedural description of the method would help people to understand things in an easier way.

---

> > ### Author Response · Authors · 2024-11-23
> > **Response to Reviewer tRL6**
> >
> > Thank you for your feedback!
> >
> > We agree that a clearer procedural description can be helpful for the reader. In response to your comment, we revised the manuscript and included a procedural description in Section 4 (lines 171-179). Specifically, we now explain that we first sample $n$ independent answers from the LLM and measure the information leakage $X_i$ in each answer. Then, we compute our (single-formula) metrics using the previously computed $X_1, \ldots, X_n$. Please note that we introduce four distinct metrics (leakage bounds, expectation bound, standard deviation bound), each offering unique insights into the leakage distribution. Your understanding is correct, each metric is essentially based on taking an average of the samples (and additionally including an error bound on the estimation).
> >
> > We hope this clear algorithmic description along with the uploaded ready-to-use code will help the reader's understanding and make the method easier to follow. Overall, we believe our probabilistic approach is an important contribution to the community, as also highlighted by other reviewers.
> >
> > We hope that we could address your feedback to your satisfaction. Please let us know if you have any additional comments or questions.

---

> > > ### Comment · Reviewer_tRL6 · 2024-11-23
> > > **Response**
> > >
> > > Thanks for the modification, and I've adjusted my score accordingly.

---

### Official Review · Reviewer_v1rM · 2024-10-31

**Soundness:** 4
**Presentation:** 3
**Contribution:** 3
**Rating:** 10
**Confidence:** 2

**Summary:**

This paper introduces a probabilistic perspective on LLM evaluation which shifts from single point estimates towards evaluating entire output distributions offers significant potential for the field of unlearning and proposes a novel framework to directly assess the output distribution of a model. Besides, an unlearning loss based on entropy optimization and adaptive temperature scaling is also proposed.

**Strengths:**

1. **Novel Perspective on LLMs Unlearning Evaluation** Existing deterministic metrics are apparently insufficient for LLM unlearning evaluation, and this paper introduces a probabilistic perspective to mitigate this issue.

2. **Adequate Mathematical Derivation** In this paper, the authors demonstrate the rationality of their method theoretically and empirically.

**Weaknesses:**

1. **More Discussion on LLM Alignment Evaluation** Since the titile contains "ALIGNMENT", more discussion on this topic should be included in this paper.

**Questions:**

See weaknesses.

---

> ### Author Response · Authors · 2024-11-19
> **Response to Reviewer v1rM**
>
> Thank you for your review!
>
> In practice, our metrics can be applied independent of the downstream task. We chose to emphasize unlearning in the main part of the paper because it is a critical area where probabilistic leakage can have particularly critical consequences, such as the inadvertent exposure of sensitive information. However, we agree that extending the discussion to alignment enhances the broader applicability of our work.
>
> In response to your comment, we provide additional alignment experiments in Appendix B to demonstrate how our probabilistic evaluation framework can be applied to alignment tasks. Specifically, we show that our proposed probabilistic metrics can seamlessly generalize to tasks beyond unlearning. These metrics require only a continuous or binary evaluation measure derived from sampling model outputs, which can then be integrated into our formulas to estimate bounds efficiently. By applying this approach to alignment, we estimate the risk of LLMs generating harmful responses, highlighting the adaptability and efficiency of our framework in alignment contexts.
>
> We hope the additional experiments and explanations in Appendix B clarify how our approach can be applied to alignment tasks and strengthen the contribution of the paper. Thank you for pointing out this opportunity for improvement!
>
> We hope that we could address all your questions to your satisfaction. Please let us know if you have any additional comments or questions.

---

> > ### Comment · Reviewer_v1rM · 2024-11-26
> >
> > Thanks for your response which have addressed my concerns, and I've adjusted my score accordingly.

---

### Official Review · Reviewer_B7r6 · 2024-11-03

**Soundness:** 3
**Presentation:** 2
**Contribution:** 3
**Rating:** 6
**Confidence:** 3

**Summary:**

This paper proposed a set of metrics that bounds the risk of unlearning by estimating the bounds of probability of leaks, and the deviation of such random variables. Instead of computing the metric over deterministic point estimates drawn from greedy decoding of LLMs, it proposes the use of Monte Carlo methods, and then estimates these bounds by computation over the empirical distribution. Additionally, the authors proposed some mitigation methods to reduce the risk of leakage when fine-tuning a LLM, and show through experiments that such measures offer potential for reducing the risk and undesired biases in model outputs.

**Strengths:**

- The paper proposes metrics that is defined on the output distribution rather than the point estimate. I think this is a remarkable step and should be considered by various other scenarios.
 - The exposition on the estimation of probability of leakage and bounds of standard deviation are intuitive and sound.
 - The set of experiments presented are convincing.

**Weaknesses:**

- Notations can be improved in the exposition. For example, $M_1, \cdots, M_4$ actually stands for estimates of different variables, rather than 4 different ways of estimating the same variable. See questions below for more suggestions.
 - Some derivation are not self-contained (e.g. in Metric 2, the $\epsilon = \sqrt{\frac{\log (1/\alpha)}{2n}}$ is not self-contained and is derived from prior work.
 - Expositions tend to be a bit too formal, and lacking some intuitions and insights. See below.

**Questions:**

- L118: $V^\infty$: I believe that the more accepted notation for sequences of arbitrary sequences is $V^*$, where $*$ is the Kleene star.
 - L150: "extend" -> "extent".
 - L177: "Binary case": This exposition feels a bit verbose. My understanding is that you are empirically fitting a Beta distribution based on whether data is leaked through your Monte-Carlo experiments, and outputting a quantile based on a desired safety level $\alpha$. Please correct me if my understanding is not correct.
 - L197: Make this part more self-contained: especially, what is the Dvoretzky-Kiefer-Wolfowitz inequality and how does it apply here?
 - L211: Proposition 3: This lower and upper bound is reminiscent of the Darboux integrals of $F_n$. If possible, please elaborate on the relationship of this bound estimate to an underlying integral expression. Additionally, it'd be good to reiterate that $F_n$ is the empirical  CDF.
 - L225: What does $\eta_i$ bound? Please discuss.
 - L230: $M_4$: I think it'd be better to call this something like $M_\sigma$, to be clear that this is not an estimate of the probability.
 - L246: $\bar X + 2\bar \sigma$: why 2? I believe this is a choice based on the safety level, but it would be better to define this as a hyperparameter whose selection is based on the accepted risk level.

---

> ### Author Response · Authors · 2024-11-19
> **Response to Reviewer B7r6**
>
> Thank you for your review!
>
> **Regarding notation (Bounds, L118, L230).** We agree with you and have revised the notation in response to your feedback. Specifically, we renamed the bounds to clarify their intended meanings: binary leakage bound $M_{bin}$, general leakage bound $M_{gen}$, expectation bound $M_\mu$, and standard deviation bound $M_\sigma$. We also followed your suggestion to use the Kleene star for denoting sequences of arbitrary length.
>
> **Regarding the binary case (L177).** Your understanding is correct. The beta distribution comes from the Clopper-Pearson confidence bound, a common method for computing  binomial confidence intervals. In response to your comment, we clarified this explanation in the manuscript.
>
> **Regarding $\epsilon$ and the DKW-inequality (L197).** Thank you for pointing out potential to make our statements more self-contained in the main paper. In response to your comment, we clarified the statements in Section 4.1 by explicitly stating and citing the DKW-inequality, showing that $\epsilon$ directly stems from it.
>
> **Regarding Proposition 3 (L211).** You are right, the bounds stem from lower and upper bounding the integral $E[X] = 1-\int F(x)$ $dx$ with Riemann sums. Please note that we elaborate on this relationship in a self-contained proof in Appendix D. In response to your comment, we have added pointers to proofs directly in the main text.
>
> **Regarding $\eta_i$ (L225).** The variable $\eta_i$ bounds the term $(x-E[X])^2$ in the variance integral. Please note that we elaborate on this in a detailed, self-contained proof in Appendix D. In response to your feedback, we  improved clarity in the main text and added pointers to the proof in the Appendix.
>
> **Regarding the ED score (L246).** We agree that generalizing the ED score would be beneficial. We revised the section in the manuscript to introduce a hyperparameter for the ED score, allowing it to be selected based on the risk level as you proposed.
>
> **Typo (L150).** We also fixed the typo in line 150, thank you for pointing this out.
>
> Please note that we provide detailed, self-contained proofs for all statements in Appendix D, offering further insights for interested readers. For readers less familiar with statistical machine learning, we also provide user-friendly and ready-to-use code uploaded as supplemental material.
>
> We hope that we could address all your questions to your satisfaction. Please let us know if you have any additional comments or questions.

---

> > ### Author Response · Authors · 2024-11-27
> > **Response to Reviewer B7r6**
> >
> > Dear reviewer, thank you again for your review. Please let us know if you have any additional comments or questions. Thank you.

---

### Official Review · Reviewer_MsxG · 2024-11-04

**Soundness:** 4
**Presentation:** 3
**Contribution:** 4
**Rating:** 10
**Confidence:** 3

**Summary:**

In this paper authors address the problem of reliable unlearning in LLMs. First they introduce a problem, that evaluations based on deterministic point estimates (sampled texts) fail to reliably catch the risks exposed in probablistic outputs. For the case of unlearning, authors state that existing methods rely on a single generated sequence to identify if the information leakage is present or not. Which might not be enough when assessed model might still eventually produce a text with leaked information (with some probability). Therefore authors propose a set of 4 metrics aiming accurately quantify information leakage in model output distribution. Then, authors propose a novel unlearning training objective, which aims to simultaneously minimize model's output distribution entropy on a set of "forget samples" while retaining diversity on "retain samples". The loss itself is a set of additional terms which can be applied to some existing unlearning objectives. Finally, authors conduct comprehensive evaluation of unlearning with different methods using the proposed metrics.

**Strengths:**

1) Authors provide 4 carefully defined metrics, along with necessary guarantee proofs. Overall, the paper is very well composed.
2) Those 4 proposed metrics allow to comprehensively evaluate model unlearning using entire output distribution (potentially, right now it is limited to MC sampling on certain examples). This appears to be a novel contribution and addresses the lack of probablistic evaluation in the field of unlearning.
3) This approach can be potentially extended to other tasks which require reliable evaluations.
4) The proposed entropy optimization objective is clearly defined and is formuated as additive terms which can be applied to existing unlearning losses, which makes it easy to implement. Addressing diversity on retain samples allows to ensure that models remains useful after unlearning.

**Weaknesses:**

1) While paper title reads as "A Probabilistic Perspective on Unlearning and **Alignment** for Large Language Models", authors effectively **do not touch** the alignment in their work, leaving it for further research. Indeed, alignment is only mentioned in Introduction, Limitations and Conclusion. This paper would benefit from having at least small discussion on how the proposed metrics can be extended to other evaluation tasks.

**Questions:**

1) The term $\alpha$, which is used extensively in formulation of metrics and overall thorough the paper, is only loosely defined in appendix. While it may be the usual practice in math-heavy papers, it does substantially confuse readers who are not so proficient. It is quite pity to read a definition or proof and find terms that simply not defined anywhere above. Consider defining $\alpha$ in the main text of the paper.
2) How increased $\lambda_r$ impacts metrics other than diversity?
3) How does proposed EO objective impacts training efficiency (in terms of increased latency or increased VRAM requirements)? Does it limit it's applicability?

---

> ### Author Response · Authors · 2024-11-19
> **Response to Reviewer MsxG**
>
> Thank you for your review!
>
> **Regarding alignment.** In practice, our metrics can be applied independent of the downstream task. We chose to emphasize unlearning in the main part of the paper because it is an area where probabilistic leakage can have particularly critical consequences, such as the inadvertent exposure of sensitive information. However, we agree that extending the discussion to alignment enhances the broader applicability of our work.
>
> In response to your comment, we provide additional alignment experiments in Appendix B to demonstrate how our probabilistic evaluation framework can be applied to alignment tasks. Specifically, we show that our proposed probabilistic metrics can seamlessly generalize to tasks beyond unlearning. These metrics require only a continuous or binary evaluation measure derived from sampling model outputs, which can then be integrated into our formulas to estimate bounds efficiently. By applying this approach to alignment, we estimate the risk of LLMs generating harmful responses, highlighting the adaptability and efficiency of our framework in alignment contexts.
>
> We hope the additional experiments and explanations in Appendix B clarify how our approach can be applied to alignment tasks and strengthen the contribution of the paper. Thank you for pointing out this opportunity for improvement!
>
> **Regarding the definition of $\alpha$ (Q1).** Thank you for pointing out potential to clarify notation. While we introduce the significance level $\alpha$ in the main text at the beginning of Section 4.1, we agree that it is a critical component for understanding the paper. In response to your comment, we added further clarifications to the manuscript (line 181).
>
> **Regarding the impact of $\lambda_r$ on metrics other than diversity (Q2).** Thanks for pointing this out. We briefly state in the caption of Figure 5 and subsection 6.3 (c) that entropy regularization has no impact on the utility of the model. We will make it more prominent in the final paper that $\lambda_r$ did not have any considerably effect on model utility or output diversity.
>
> **Regarding the proposed entropy optimization objective (Q3).** For entropy optimization we simply compute the entropy of the softmax output of the model. This loss computation has no measurable effect on the speed of the training in our experiments.
>
> We hope that we could address all your questions to your satisfaction. Please let us know if you have any additional comments or questions.

---

### Author Response · Authors · 2024-11-19
**Global response**

We want to thank all reviewers for their valuable feedback. We changed our initial submission in response to their comments as follows:

- Reviewers MsxG, v1rM and tRL6: We have included additional alignment experiments in Appendix B to illustrate that our probabilistic evaluation framework generalizes to alignment tasks.
- Reviewer tRL6: Clarifications on the algorithmic procedure of our probabilistic evaluation framework.
- Reviewer B7r6: Improved notation and few clarifications.

---

### Meta-Review · Area_Chair_GKXw · 2024-12-18

**Metareview:**

This paper proposes a formal probabilistic evaluation framework for LLMs and designs new metrics with high-probability guarantees concerning the output distribution of a model. In addition, an unlearning loss based on entropy optimization and adaptive temperature scaling is also proposed. The proposed framework is novel and important. The evaluation results show the effectiveness of the proposed metrics and method. The paper is generally well written. There are some minor issues with the presentation that need to be addressed in the final version.  All reviewers ultimately have a positive opinion of the paper.

**Additional Comments On Reviewer Discussion:**

Two reviewers increased their scores during the rebuttal period and all reviewers ultimately have a positive opinion of the paper.

---

### Decision · Program_Chairs · 2025-01-22

Accept (Oral)